# Hidradenitis Suppurativa: An Understanding of Genetic Factors and Treatment

**DOI:** 10.3390/biomedicines12020338

**Published:** 2024-02-01

**Authors:** Yi-Lun Chu, Sebastian Yu

**Affiliations:** 1Department of Dermatology, Kaohsiung Medical University Hospital, Kaohsiung 807377, Taiwan; h60733@gmail.com; 2School of Post-Baccalaureate Medicine, College of Medicine, Kaohsiung Medical University, Kaohsiung 807378, Taiwan; 3Department of Dermatology, College of Medicine, Kaohsiung Medical University, Kaohsiung 807378, Taiwan; 4Neuroscience Research Center, Kaohsiung Medical University, Kaohsiung 807378, Taiwan; 5Master of Public Health Degree Program, National Taiwan University, Taipei 100025, Taiwan

**Keywords:** hidradenitis suppurativa, inflammatory skin disease, NCSTN, γ-secretase, TNF-α, IL-17, IL-1β, IL-12, IL-23, targeted therapy, JAK inhibitor

## Abstract

Hidradenitis suppurativa (HS), recognized as a chronic and debilitating skin disease, presents significant challenges in both diagnosis and treatment. This review explores the clinical manifestations, genetic landscape, and molecular mechanisms underlying HS. The disease’s association with a predisposing genetic background, obesity, smoking, and skin occlusion underscores the complexity of its etiology. Genetic heterogeneity manifests in sporadic, familial, and syndromic forms, with a focus on mutations in the γ-secretase complex genes, particularly NCSTN. The dysregulation of immune mediators, including TNF-α, IL-17, IL-1β, and IL-12/23, plays a crucial role in the chronic inflammatory nature of HS. Recent advancements in genetic research have identified potential therapeutic targets, leading to the development of anti-TNF-α, anti-IL-17, anti-IL-1α, and anti-IL-12/23 therapies and JAK inhibitors. These interventions offer promise in alleviating symptoms and improving the quality of life for HS patients.

## 1. Introduction

Hidradenitis suppurativa (HS), also known as acne inversa (AI), is a chronic, recurrent, debilitating skin disease [1] thought to involve occlusion of the hair follicle at the pilosebaceous unit (PSU) such as axillary, inguinal, and anogenital regions [2,3]. Clinically, HS manifests as painful inflammatory nodules, abscesses, and interconnected tunnels emitting malodorous discharge and results in disfiguring scarring that has a considerable impact on a patient’s quality of life [4,5,6]. Similarly, steatocystoma multiplex, an autosomal dominant disease that can lead to the formation of suppurative syringomas, shares similar characteristics [7]. The approximate occurrence rate of the condition varies from 1% to 4%. The age group with the highest rates of incidence is young adults falling between 20 and 40 years, with rates peaking at 4% [8,9]. The onset of HS is notably associated with risk factors, with a predisposing genetic background, obesity, smoking, and skin occlusion being identified as significant contributors, suggesting that this skin condition could be influenced by both the surroundings a person is exposed to and their personal habits and choices [10]. 

Due to the genetic heterogeneity and complexity involved, HS can be classified into three distinct forms: sporadic, familial, and syndromic [11]. While HS typically presents as a sporadic disease, there are instances where it manifests as a familial disorder. In approximately 40% of cases, a familial form is observed, characterized by an autosomal dominant mode of inheritance. In cases where there is a family history of HS, the affected individuals are more likely to be female. Additionally, this familial HS is characterized by severe and widespread symptoms. Clinically, these cases can be identified by an earlier age at onset, and the condition often affects the axillary region [12]. For another small group of patients, HS can be associated with other immune-mediated inflammatory diseases or inherited conditions, presenting as “syndromic” HS [13].

The prompt diagnosis and early treatment of HS are crucial for symptom management and minimizing the formation of new lesions. However, the mean time to diagnose HS is prolonged, averaging seven years [14], due to the diverse presentation of the disease in various phenotypes, which can make diagnosis challenging and confusing. HS is primarily diagnosed clinically, and unlike other conditions related to HS, such as acne conglobata or dissecting cellulitis, a biopsy is typically unnecessary to establish the diagnosis. However, procedures like skin biopsy, bacterial culture, and ultrasound may be employed to rule out alternative diagnoses [15,16]. In cases where there is uncertainty, these diagnostic tools can help exclude other conditions. Three criteria collectively contribute to the accurate diagnosis of HS, allowing for timely intervention and management [17]. First, early lesions should exhibit deep-seated painful nodules, often described as “blind boils” without a purulent point, and secondary lesions should include abscesses, sinuses, bridged scars, and “tombstone” pseudocomedones. Second, lesions must manifest in at least one characteristic body area, typically involving the axillae, groin, perineal and perianal region, buttocks, and inframammary and intermammary folds. Additionally, the distribution is frequently bilateral. Third, the diagnosis is further supported by the chronic nature of the disease, marked by persistent relapses and recurrences. Moreover, medical history also serves as a valuable diagnostic tool. Key indicators include the onset of symptoms during adolescence or young adulthood, a history of recurrent or persistent disease, and any family history of HS [18].

In clinical practice, the Hurley system is used for the staging of HS [18]. Patients with the disease are categorized into three disease severity groups. In Stage I, individuals experience the formation of solitary or multiple abscesses without sinus tracts or scarring. Stage II is characterized by recurrent abscesses with sinus tracts and scarring involving single or multiple separated lesions. For patients classified into Stage III, there is diffuse or broad involvement, with multiple interconnected sinus tracts and abscesses across their entire area. 

Progress has been achieved in understanding the pathological processes contributing to the development of HS. Factors such as genetic and environmental influences, lifestyle choices, hormonal status, and the microbiota collectively play a role in triggering immune activation around hair follicles and causing hyperkeratosis in the infundibulum [19]. These changes lead to follicular plugging and stasis. Bacterial growth in skin folds typically intensifies immune activation. Immune cells, including innate (e.g., macrophages, granulocytes) and adaptive immune systems (e.g., T helper [TH] 1 cells, TH17 cells), release proinflammatory cytokines to activate tissue cells and further promote immune cell infiltration and inflammation [20]. Despite commonly observed clinical features like bacterial infection and follicular hyperkeratosis, the precise molecular mechanisms underlying HS remain to be fully understood [13]. This review will discuss the mechanism of HS based on the genetic and molecular pathogenesis that activates the immune system. We also aim to provide biological therapies available for HS.

## 2. Search Strategy

We conducted a systematic search on PubMed to locate original articles presenting findings on the genetic factors associated with HS and its current biologic agents. Our search strategy involved utilizing various combinations of free-text keywords and Medical Subject Heading (MeSH) terms, including but not limited to “Hidradenitis Suppurativa”, “inflammatory skin disease”, “Genetics”, “Cytokines”, “mechanism”, “pathogenesis”, and “antagonists and inhibitors.” Boolean operators such as “AND” and “OR” were employed to refine the search. Specific biologic agents’ names were also incorporated into the search terms. Additionally, we scrutinized the references of clinical trials, review articles, and observational studies that delved into the mechanism of HS. Articles not in English or those addressing different forms of HS were excluded from our analysis.

## 3. Genetics of Hidradenitis Suppurativa

Genomic research on HS has been ongoing to better understand the genetic factors and molecular mechanisms underlying the condition. The first research on genetic factors in HS was published by Fitzsimmons and Guilbert in 1985. They clarified the evidence of autosomal dominant inheritance or familial HS [21]. In 2006, Min Gao et al. suggested 1p21.1–1q25.3 as a possible HS locus, which was the first causative gene for HS and represents a starting point for understanding the molecular mechanisms of this disease [22]. Later, in 2011, Min Gao et al. further confirmed the gene *NCSTN* responsible for HS. Their research successfully introduced a valuable and efficient approach for identifying the causative gene of a rare monogenic disorder, even in small sample sizes. Additionally, it contributed to advancing our comprehension of the genetic underpinnings of HS [23]. Subsequently, multiple researchers have presented evidence that bolsters the idea of a monogenic origin in familial HS, with a predominant focus on loci responsible for encoding proteins within the γ-secretase complex [2]. Investigations have concentrated on sequencing the crucial subunits of γ-secretase in patient cohorts with HS, revealing a total of 20 reported mutations in *NCSTN* thus far [24]. These initial findings have spurred a determined effort to delve into the role of genetic susceptibility in HS, aiming to enhance our understanding of the disease’s pathophysiology.

In addition to γ-secretase gene mutation, researchers also tried to elucidate the immunologic landscape of HS that can lead to chronic inflammation. Numerous studies demonstrate elevated levels of proinflammatory cytokines (cytokines associated with hidradenitis suppurativa are listed in Table 1). In 2009, Matusiak, L., Bieniek, A., and Szepietowski, J.C., revealed that levels of TNF-α serum concentration are higher in HS individuals than in healthy controls, which is the first report of increased TNF-α serum concentration in HS patients [25]. Elevated levels of IL-1β in HS skin were first described by van der Zee, H.H. et al. in 2011 using two distinct cytokine bead arrays to measure the difference in a spectrum of inflammatory cytokines between HS lesions and healthy control skin samples [26]. Over the past two decades, IL-17 cytokines have emerged as significant contributors to various inflammatory disorders, demonstrating particular prominence in skin inflammation [27]. The substantial expression of IL-12/Th1 and IL-23/Th17 pathways further underscores the involvement of the immune system in HS [28]. Research by Schlapbach, C. et al. revealed abundant expression of IL-12 and IL-23 by macrophages infiltrating both the papillary and reticular dermis of lesioned skin [29]. This section begins with a brief review of the history of research. The molecular mechanism and signaling pathways of these genes and cytokines and how they influence our immune system will be discussed in the forthcoming sections.

### 3.1. γ-Secretase Gene

γ-secretase is an endoprotease complex responsible for numerous type-1 transmembrane proteins that undergo intramembranous cleavage [30]. It is a high-molecular-weight complex minimally composed of four components: presenilins (*PSEN*), nicastrin (*NCSTN*), anterior pharynx defective 1 (*APH1A*), and presenilin enhancer 2 (*PSENEN*) [31]. There are six genes encoding proteins for the γ-secretase complex, three of which have been reported mutated in HS (*NCSTN*, *PSENEN*, and *PSEN* [1]), which may interrupt the roles played by the type-1 transmembrane proteins like cell signaling pathway and cellular adhesion [32].

The Notch signaling pathway is a highly conserved mechanism that participates and plays crucial roles not only in intercellular communication but also in embryogenesis and homeostasis [33]. Notch is a type of type I transmembrane protein that plays a role as a surface-embedded receptor and binds ligands on neighboring cells. Ligand binding leads to cleavage at site 2 (S2) first by a disintegrin and metalloprotease 10 (ADAM10) to remove the extracellular domain of Notch protein. Then, its transmembrane domain will be cleaved by γ-secretase at site 3 (S3) to release the Notch intracellular domain (NICD) [34], which is then freed and migrates into the cell nucleus. NICD then interacts with the transcriptional cofactors DNA binding protein CBF1—Suppressor of Hairless—LAG1 (CSL), p300, and co-activator mastermind-like transcriptional co-activator 1 (MAML1) to activate transcription about regulation of cell proliferation and survival [33]. In HS patients with *NCSTN*, *PSENEN*, or *PSEN* mutation, their downstream protein Nicastrin, Presenilin enhancer 2, or Presenilin is absent or defective. The impaired γ-secretase function makes NCID unable to be cleaved; thus, it cannot translocate into the nucleus to regulate gene transcription, leading to abnormal follicular keratinization, epidermal hyperplasia, cyst formation, and absence of sebaceous glands [35]. 

Besides Notch signaling, the blocking of *NCSTN* may also lead to the impairment of phosphatidylinositol 3-kinase (PI3K)/protein kinase B (AKT) and the epidermal growth factor receptor (EGFR) pathway. A microRNA, *miR-100-5p*, is found to be downregulated in both familial HS and Nicastrin-knocked out mice. Although it is believed to participate in the AKT signaling pathway, the relationship between HS with *NCSTN* mutation and the AKT signaling pathway still remains to be proved [36]. The level of another microRNA, *miR-30a-3p*, is found to be decreased in HS individuals with *NCSTN* mutation. As it negatively regulates the expression of RAB31, a Ras-related protein, the levels of RAB31 increase and accelerate the degradation of activated EGFR, contributing to abnormal differentiation of keratinocytes [37].

### 3.2. TNF-α

In HS individuals, the level of TNF-α serum concentration is found to be significantly higher than in the sera of healthy controls [26]. TNF-α is a cytokine synthesized and secreted by macrophages, T lymphocytes, and natural killer cells whose structure is homotrimer with 157 amino acids. This cytokine has two forms: soluble (sTNF-α) and transmembrane (tmTNF-α). The transmembrane form is synthesized as a precursor and is transferred to soluble form by TNF-α-converting enzyme (TACE), a membrane-bound disinterring metalloproteinase. Both forms bind to type 1 receptors (TNFR1), while type 2 receptors (TNFR2) are mainly combined with tmTNF-α. The differences between TNFR1 and TNFR2 are that the former exists in all human tissues and has a death domain, whereas the latter is expressed in immune cells only with no death domain [38].

Once sTNF-α or tmTNF-α binds to TNFR1, trigger complex I, IIa, IIb, and IIc will be activated. TNFR1 will recruit and combine with TNFR1-associated death domain (TRADD), receptor-interacting serine/threonine-protein kinase 1 (RIPK1), TNFR-associated factor 2 or 5 (TRAF2/5), cellular inhibitor of apoptosis protein 1 or 2 (cIAP1/2), and linear ubiquitin chain assembly complex (LUBAC) to facilitate complex I assembling [39]. The assembled complex I activates nuclear factor κB (NF-κB) and mitogen-activated protein kinases (MAPKS) signaling pathway, leading to inflammation and immunological reaction to defend pathogens [40,41]. Trigger complex IIa consists of TRADD, RIPK1, TRAF2, cIAP1/2, pro-caspase 8, and Fas-associated protein with death domain (FADD). It forms together with complex IIb, which has the same composition as complex IIa with the addition of RIPK3, to become an apoptosome that activates caspase 8, thus inducing apoptosis [42]. Complex IIc, also known as necrosome, comprises RIPK1 and RIPK3. It activates mixed lineage kinase domain-like protein (MLKL) via RIPK3-mediated phosphorylation, ultimately leading to necroptosis and inflammatory reaction [43].

TNFR2 recruit TRAF1/2 and cIAP1/2 to assemble together to activate NF-κB, MAPKS, and AKT signaling pathways to play a critical role in homeostatic bioactivities [44]. In summary, TNFR1 can be viewed to participate in cytotoxic and proinflammatory functions, and TNFR2 can be considered responsible for the regulation of cell activation, migration, and proliferation. A trigger in either receptor could contribute to the pathogenesis of chronic inflammatory diseases like HS.

### 3.3. IL-17

IL-17, a cytokine that plays a key role in multiple inflammatory disorders, is produced by Th17 cells [45]. IL-23 binds to receptors on CD4+ Th17 cells, inducing the phosphorylation of Janus kinase (JAK) and receptor tyrosine kinase (RTK). This phosphorylation induces signal transducer and activator of transcription proteins 3 (STAT3) to assemble into a homodimer, which then regulates the expression of a transcription factor, retinoid-related orphan receptor-γt (ROR-γt), to produce IL-17A and IL-17F [46]. Both IL-17A and IL-17F bind to IL-17 receptor (IL-17R) A, C, and D on the surface of keratinocytes to facilitate its proliferation [47]. After binding to ligands, IL-17R acts on the Act 1 adaptor protein (Act1) to induce the ubiquitination of TNF receptor-associated factor 6 (TRAF6) [48,49]. The ubiquitinated TRAF6 recruits transforming growth factor-β-activated kinase 1 (TAK1) and inhibitor of kappa B kinase (IKK) complex and then promotes the activation of NF-κB signaling pathway to induce an inflammatory reaction. In another way, the IL-17R plus Act1 complex will further combine with TRAF4, mitogen-activated protein kinase kinase kinase 3 (MEKK3), and mitogen-activated protein kinase kinase 5 (MEK5) to activate extracellular signal-regulated kinase 5 (ERK5), contributing to the proliferation of keratinocytes [50].

### 3.4. IL-1β

IL-1 is a proinflammatory cytokine produced by macrophages during immune reactions, which helps defend against pathogens and physical injuries. There are two individual forms of IL-1, IL-1α and IL-1β. They have similar biological functions, although encoded in two distinguished cDNAs [51]. After macrophages sense pathogen-associated molecular pattern molecules (PAMPs), damage-associated molecular pattern molecules (DAMPs), or environmental stress such as trauma, ischemia, and tissue damage [52], NLR family pyrin domain-containing 3 (NLRP3) inflammasome, an intracellular multiprotein complex is formed [53]. IL-1β is initially produced as a precursor that needs to be cleaved by caspase-1, caspase-8, elastase, chymase, and proteinase 3 in the NLRP3 inflammasome to become biologically active [54]. Various factors, including exogenous agents and endogenous particles, can induce the release of bioactive IL-1β [55]. IL-1β binds to type I IL-1 receptor (IL-1R1) with a co-receptor called IL-1R accessory protein (IL-1RAcP) [56]; this trimeric complex recruits and assembles with myeloid differentiation factor 88 (MYD88) and interleukin 1 receptor-associated kinase 4 (IRAK4) to form the first stable molecule during IL-1 signaling transduction process [57]. TRAF6, a ubiquitin E3 ligase, is then recruited and oligomerized after the series of phosphorylation of IRAK. Oligomerized TRAF6 attaches K63-linked polyubiquitin chains to IRAK1, TGF-β–activated protein kinase-binding protein 2 (TAB2), TAB3, and TAK-1 with ubiquitin E2 ligase [56]. Then, the ubiquitinated TAK-1 facilitates the association between TRAF6 and MEKK3 [58,59], thus activating NF-κB, c-Jun N-terminal kinases (JNK), and p38 MAPK pathways to induce the expansion of naïve and memory CD4+ T cells as well as promote keratinocyte proliferation [60].

### 3.5. IL-12/23

Upon pathogen invasion of the human body, antigen-presenting cells (APCs), such as macrophages and dendritic cells, secrete cytokines to activate our immune system. IL-12 released by APCs triggers the differentiation of naïve T cells into Th1 cells [61]. On the other hand, IL-6 and TGF-β1 induce naïve T cells to become Th17 cells [62,63]. Th1 cells secrete IL-12, whereas Th17 cells secrete IL-23; both of the cytokines serve as a bridge between innate and adaptive immunity. IL-12 promotes Th1 cells to proliferate and produce interferon-γ, a cytokine that participates in defending against viruses and parasites through tyrosine kinase 2 (TYK2) and STAT4 phosphorylation [64]. As for IL-23, a member of the IL-12 cytokine family, it is a heterodimeric cytokine composed of IL-12p40 and IL-23p19; they bind to IL-12Rβ1 and IL-23Rα on the membrane of Th17 cells, respectively [65,66]. After ligand binding, IL-17 is produced to participate in inflammation and keratinocyte proliferation via JAK2, STAT3, and RORγt pathways [45,46]. Thus, the abnormal mechanism can contribute to pathological changes in HS. Also, in addition to T-helper cells, IL-23 and IL-12 can be produced by monocytes and dendritic cells, and the balance between them is regulated by prostaglandin E2 (PGE2) [67]. PGE2 plays a role in an anti-inflammatory manner by decreasing IL-23 production by monocytes, while it has the opposite effect in dendritic cells [68]; thus, the loss of control may lead to inflammation.

## 4. Current and Potential Therapeutic Agents Targeting Immune Mediators in Hidradenitis Suppurativa 

### 4.1. Anti-TNF-α Therapy

Currently, there are some TNF-α inhibitors for HS. 

In 2015, the FDA approved the first biotechnology drug for HS: adalimumab. It is a fully human recombinant anti-TNF-α immunoglobulin (Ig) G1 monoclonal antibody that binds and neutralizes sTNF-α and tmTNF-α to inhibit the inflammatory cascade leading to HS skin lesions [69,70]. The administration of adalimumab involves an initial subcutaneous dose of 160 mg, followed by an 80 mg dose after 2 weeks, and subsequently, a maintenance dose of 40 mg weekly [60]. Although substantial evidence supports its efficacy and safety [71], patients who are younger than four years old and weigh less than 15 kg are not advised to use it due to there being no adequate study about its safety for this group.

Infliximab is a chimeric IgG monoclonal antibody protein derived from recombinant DNA, incorporating components from both murine and human sources. Despite its off-label use to treat HS, it targets TNF-α and also has presumed efficacy [72]. For patients who received an initial dose of 7.5 mg/kg followed by a maintenance frequency every 4 weeks, 47.6% at week 4 and 70.8% at week 12 achieved a clinical response. If those patients have an incomplete initial response, the dosage escalates to 10 mg/kg, which can achieve a clinical response of 37.5% at week 4 and 50% at week 12 [73].

Etanercept, an FDA-approved anti-inflammatory agent for several autoimmune diseases, is a fusion protein consisting of the p75-Fc region of the human TNF-α receptor. It functions by competitively attaching to membrane-bound TNF-α receptors [74]. Although some small case series and an open-label phase II trial illustrated the efficacy of etanercept in early improvement of disease activity and local pain when given 50 mg once a week subcutaneously [75,76,77], it failed to show any benefit in other clinical trials [78]. So far, there is no adequate evidence to support the benefit and optimal dosage of etanercept in HS.

Certolizumab is a recombinant humanized IgG4 that contains only polyethylene glycol antigen-binding fragments. Without a fragment-crystallizable region, this agent cannot pass through the placenta [74], indicating that usage on pregnant individuals is relatively acceptable. Six case reports demonstrated the effectiveness of certolizumab in treating HS patients in which certolizumab was given 200 or 400 mg every other week [79]. However, it is not yet sanctioned for treating HS. The effectiveness and safety of certolizumab for HS treatment remain inadequately established, warranting further exploration through larger-scale trials.

Regarding golimumab, another monoclonal anti-TNF-α antibody that binds both forms of TNF-α, it has been under study for its efficacy and safety in treating patients with HS. One case report demonstrated that a patient received a subcutaneous dose of 200 mg, followed by 100 mg subcutaneously every 4 weeks, combined with antibiotic therapy, resulting in the disappearance of HS lesions with no recurrence [80]. In another case, a dosage of 50 mg subcutaneously once a month for 8 months showed no benefits in improving the disease [81]. This suggests that more research is needed to confirm the effectiveness of this drug in HS treatment, and a higher dosage might be necessary for the treatment of HS.

### 4.2. Anti-IL-17 Therapy

Secukinumab is a fully human monoclonal antibody designed to specifically target IL-17A. It functions by inhibiting the binding of IL-17A with IL-17R, consequently impeding keratinocyte hyperproliferation and T-cell infiltration and thus attenuating the immune response. Though currently used off-label for HS [69], two phase 3 trials, SUNRISE and SUNSHINE, showed its efficacy in patients with moderate-to-severe HS. It is effective at rapidly improving clinical presentation with a favorable safety profile when given as a subcutaneous dosage 300 mg every 2 weeks [82].

Instead of targeting cytokines, brodalumab is a fully human monoclonal IgG2 antibody that inhibits the function of IL-17 by directly binding to its receptor, IL-17RA [83]. By acting as an antagonist, it inhibits interactions with cytokines and is thus explored as an off-label therapy for HS. An open-label cohort study administered brodalumab to ten patients with a dose of 210 mg/1.5 mL subcutaneously at weeks 0, 1, and 2 and every 2 weeks until week 24. All of the ten patients achieved hidradenitis suppurativa clinical response (HiSCR) with no serious adverse effects [84].

Unlike secukinumab, which targets IL-17A only, bimekizumab, an IgG1 monoclonal antibody, is created to inhibit both IL-17A and IL-17F driving inflammatory processes [85]. The data of BE HEARD I and BE HEARD II phase 3 studies were presented recently by UCB, a biopharmaceutical company, revealing that bimekizumab achieved substantial and clinically significant responses in patients with HS. Over 55% of patients achieved HiSCR50 by week 16. Until week 48, over 60% of patients attained HiSCR75, and around 30% reached HiSCR100.

Ixekizumab has emerged as a prominent subject in recent studies exploring its efficacy in treating HS. As a monoclonal antibody belonging to the IgG4 class, it has been humanized to target and neutralize both soluble IL-17A and IL-17 A/F [86]. Presently, several case reports highlight its effectiveness. Patients in these studies received a dosage of 60 mg initially, followed by 80 mg at weeks 2, 4, 6, 8, 10, and 12. In one of the reports, 80% of patients achieved HiSCR at week 12 [87]. Additional case reports also demonstrated a substantial improvement in the disease, suggesting that ixekizumab holds promise as a potential therapeutic agent for HS. However, further research is imperative to validate these findings.

Also, two novel nanobody drugs, sonelokimab and izokibep, are in phase II clinical trials to evaluate their efficacy in patients with moderate to severe HS [88]. 

### 4.3. Anti-IL-1 Therapy

Anakinra, a recombinant human IL-1Ra, acts as a receptor antagonist by competitively suppressing the binding of both IL-1α and IL-1β to their receptors [86]. It is primarily employed in the treatment of rheumatoid arthritis. Two studies have investigated its impact on HS patients. In an open-label study, six patients underwent anakinra therapy, resulting in a significant mean decrease in their modified Sartorius score at week 8 among five patients [89]. Another double-blind, randomized clinical trial involved 20 patients divided into two groups: one was prescribed a subcutaneous 100 mg dose of anakinra, and the other received a placebo once daily. Following a 12-week follow-up, 78% of patients receiving anakinra showed a reduction in disease activity scores compared to 20% in the placebo group. Additionally, there was a decrease in the production of interferon-γ and an increase in the production of interleukin 22 [90]. These findings indicate that anakinra holds promise as a therapeutic agent beneficial to HS patients.

Canakinumab, a human monoclonal antibody specified for the blockage of IL-1β signal, has obtained approval from the US FDA for treating familial cold auto-inflammatory syndrome and Muckle–Wells syndrome. Numerous clinical studies have established its potential therapeutic benefits in conditions such as rheumatoid arthritis, systemic-onset juvenile idiopathic arthritis, and gout arthritis [91]. However, as of now, the evidence supporting its effectiveness in HS patients is limited to case reports and series, where it was administered as a 150 mg subcutaneously dose once a week. Further research is required to thoroughly investigate and confirm its efficacy in the context of HS treatment.

As IL-1α and IL-1β share similar physiological functions, biopharmaceuticals targeting IL-1α have been off-labeled and used for HS. A phase II open-label study of bermekimab indicated its safety and efficacy in patients with moderate-to-severe HS. Bermekimab is a true human monoclonal antibody that has a high affinity with IL-1α. It was administered subcutaneously at a dose of 400 mg per week for 13 weeks to two groups of patients with naïve or failed anti-TNF therapy previously. After 12 weeks of treatment, 61% and 63% of the two groups, respectively, achieved HiSCR. Significant reductions in abscesses and inflammatory nodules, as well as in patients experiencing pain, were seen in both groups, with the only adverse effect being injection site reactions [92]. 

### 4.4. Anti-IL-12/23 Therapy and Anti-IL-23 Therapy

Ustekinumab is a human monoclonal IgG1 antibody employed for the management and treatment of various inflammatory conditions. This antagonist blocks the p40 subunit of IL-12 and IL-23, thus suppressing the interaction of these cytokines with the IL-12Rβ1 receptor on the surface of natural killer cells and T cells. In doing so, ustekinumab can result in downregulation of the immune system [93]. A case series enrolled ten patients to evaluate the therapeutic outcomes of ustekinumab. Improvements in the Physician Global Assessment Score and Numerical Pain Rating Scale were observed in 70% and 80% of patients, respectively, with no severe adverse effects reported [94].

Guselkumab is a human IgG1λ monoclonal antibody that selectively targets the p19 subunit of IL-23, thus blocking the IL-23-mediated signaling pathway [95]. A phase IIa trial was carried out for patients with moderate-to-severe HS [96]. Twenty patients were subcutaneously administered guselkumab 200 mg every 4 weeks for 16 weeks. A total of 65% of patients achieved HiSCR at the end of the trial, with a statistically significant decrease in the median international hidradenitis suppurativa severity score system (IHS4) score. However, a lower HiSCR response of 45–50.8% was reported in another phase IIb NOVA trial, indicating that guselkumab seems only to be beneficial to a minor group of HS patients [96].

Risankizumab, another humanized IgG1 monoclonal antibody that impedes immune signaling by targeting the p19 subunit of IL-23 [97], was investigated for its efficacy and safety in patients with moderate-to-severe HS. A phase II trial allocated patients randomly to receive risankizumab 180 mg, risankizumab 360 mg, or placebo subcutaneously. HiSCR was achieved by 46.8%, 43.4%, and 41.5% at week 16, respectively, indicating that risankizumab does not appear to be an effective treatment for moderate-to-severe HS [98].

### 4.5. Janus Kinase Inhibitors

As the JAK pathway plays a crucial role in the pathophysiology of HS, the blockade of this pathway is considered to offer a novel treatment option for HS. INCB054707 is an example that has undergone two clinical trials to assess its efficacy and safety in the treatment of HS. In these trials, 10, 9, 9, 8, and 9 patients who received INCB054707 at doses of 15, 30, 60, 90 mg, or placebo, respectively, once daily, achieved HiSCR at rates of 43%, 56%, 56%, 88%, and 57% at week 8. Although improvement in life quality, IHS4, and skin pain was observed, several patients experienced treatment-emergent side effects, including upper respiratory tract infection and thrombocytopenia [99]. 

Tofacitinib and upadacitinib are two additional JAK inhibitors, and a case report highlights the potential benefits of tofacitinib in HS. In this report, two patients received a dosage of 5 mg twice daily in conjunction with other medications, such as antibiotics and immunosuppressants [100]. Moreover, an ongoing phase II trial, including 68 patients, is currently assessing the efficacy of upadacitinib in treating individuals with moderate to severe HS. Prescribed doses include upadacitinib 30 mg orally or a placebo, followed by upadacitinib 15 mg. The primary outcome is the percentage of participants achieving HiSCR at Week 12.

Furthermore, various drugs are currently undergoing clinical trials for HS treatment, including PF 06650833, PF 06700841, PF 06826647, PF-06650833, brepocitinib, ropsacitinib, and KT-474 [86].

The current and potential therapeutic agents and their targeting immune mediators are summarized in Figure 1.

## 5. Conclusions

In conclusion, this review provides a thorough examination of the multifaceted aspects of HS, elucidating its clinical presentation, genetic heterogeneity, and molecular mechanisms. The intricate interplay between genetic factors, particularly mutations in the γ-secretase complex genes such as *NCSTN*, and dysregulation of immune mediators such as TNF-α, IL-17, IL-1β, and IL-12/23 contributes to the chronic inflammatory nature of HS. As our understanding of the genetic and molecular landscape of HS advances, targeted therapeutic approaches have emerged, including anti-TNF-α, anti-IL-17, anti-IL-1α, and anti-IL-12/23 therapies and JAK inhibitors. These interventions show promise in alleviating symptoms and improving the quality of life for individuals with HS. Further research and clinical trials are warranted to optimize treatment strategies and enhance our grasp of the intricate pathogenesis of this debilitating skin disorder.

## Figures and Tables

**Figure 1 biomedicines-12-00338-f001:**
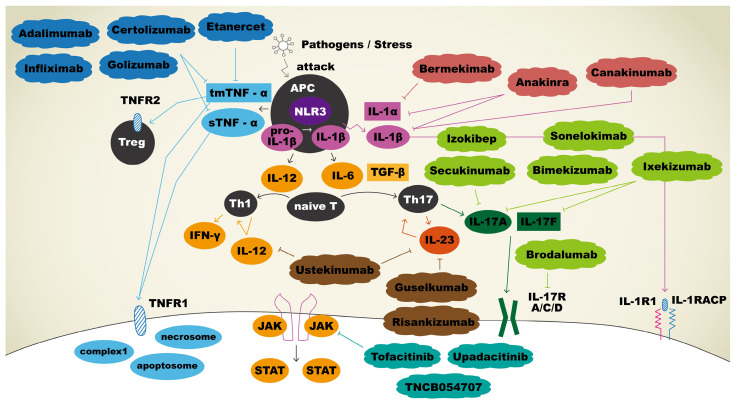
Immunopathogenesis and therapeutic agents targeting immune mediators in hidradenitis suppurativa. Upon APCs sensing pathogens, tmTNF-α and sTNF-α were secreted, and both bound to TNFR1 to facilitate apoptosis, necroptosis, and inflammation reactions. tmTNF-α will also bind to TNFR2 to play a role in regulation of cell activation, migration, and proliferation. Adalimumab, certolizumab, golimumab, and infliximab bind and neutralize sTNF-α and tmTNF-α, thus inhibiting the inflammatory cascade. IL-1β is initially produced as a precursor and then cleaved by NLR3 to become a mature form. Mature IL-1β binds IL-1R1 with co-receptor IL-1RAcP to promote keratinocyte proliferation. Anakinra inhibits both IL-1α and IL-1β, whereas sonelokimab targets IL-1β, and bermekimab targets IL-1α specifically to use off-label for HS. APCs release IL-12 for the differentiation of naïve T cells into Th1 and IL-6 for the differentiation of naïve T cells into Th17 cells. Th17 cells produce IL-23, which promotes the production of IL-17. IL-17 binds to its receptor, contributing to the proliferation of keratinocytes. Sekukinumab, bimekizumab, izokibep, sonelokimab, ixekizumab, and brodalumab are biologic agents that block IL-17 pathway and thus have been explored as off-label treatments for HS. Ustekinumab is an antagonist that blocks both IL-12 and IL-23 to result in the downregulation of immune system. Guselkumab and risankizumab bind selectively to IL-23 only. Three biologics act as JAK inhibitors: tofacitinib, upadacitinib, and TNCB054707.

**Table 1 biomedicines-12-00338-t001:** Cytokines associated with hidradenitis suppurativa and the reaction they induced.

Cytokine	Receptor	Activated Pathway	Induced Reaction
TNF-α	TNFR1 TNFR2	NFκB, MAPKS, Caspase8, MLKL NFκB, MAPKS, AKT	Cytotoxic and proinflammation Cell activation, migration, proliferation
IL-17	IL-17R A, C, D	NFκB MEK5	Inflammation Keratinocyte proliferation
IL-1β	IL-1R1 Co-receptor: IL-1RAcP	NFκB, JNK, p38 MAPK	Naïve T-cell and CD4+ memory T-cell expansion Keratinocyte proliferation
IL-12	IL-12R (IL-12Rβ1 + IL-12Rβ2)	TYK2, STAT4	Th1 proliferation and TFN-γ production
IL-23	IL-12Rβ1 + IL-23Rα	JAK, RTK, STAT, ROR-γt	Th17 release IL-17

TNF-α: tumor necrosis factor-α. TNFR: tumor necrosis factor receptor. NFκB: nuclear factor κB. MAPKS: mitogen-activated protein kinases. MLKL: mixed lineage kinase domain-like protein. IL-17R: IL-17 receptor. MEK5: mitogen-activated protein kinase kinase 5. IL-1R1: type I IL-1 receptor. IL-1RAcP: IL-1R accessory protein. JNK: c-Jun N-terminal kinases. IL-12R: IL-12 receptor. TYK2: tyrosine kinase 2. STAT: signal transducer and activator of transcription proteins. Th1: T helper 1 cell. TFN-γ: interferon-γ. IL-23R: IL-23 receptor. JAK: Janus kinase. RTK: receptor tyrosine kinase. ROR-γt: retinoid-related orphan receptor-γt. Th17: T helper 17 cell.

## Data Availability

No new data were created or analyzed in this study. Data sharing is not applicable to this article.

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
