# Peer review of "Hidradenitis Suppurativa: An Understanding of Genetic Factors and Treatment"

_biomedicines, 2024, doi:10.3390/biomedicines12020338_

Round 1
Reviewer 1 Report
Comments and Suggestions for Authors
Dear Authors
The manuscript aims to provide a comprehensive overview on the genetic background and treatment of HS. The paper is logically structured and well-written.
In my opinion, some drugs currently used off-label or under development are missing.
You can look at recent reviews on the topic and integrate:
-Świerczewska Z, Lewandowski M, Surowiecka A, Barańska-Rybak W. Immunomodulatory Drugs in the Treatment of Hidradenitis Suppurativa-Possibilities and Limitations. Int J Mol Sci. 2022 Aug 26;23(17):9716. doi: 10.3390/ijms23179716. PMID: 36077114; PMCID: PMC9456321.
- Ocker L, Abu Rached N, Seifert C, Scheel C, Bechara FG. Current Medical and Surgical Treatment of Hidradenitis Suppurativa-A Comprehensive Review. J Clin Med. 2022 Dec 6;11(23):7240. doi: 10.3390/jcm11237240. PMID: 36498816; PMCID: PMC9737445.
Author Response
Response to reviewer 1
The manuscript aims to provide a comprehensive overview on the genetic background and treatment of HS. The paper is logically structured and well-written.
Thank you for your positive feedback regarding our manuscript. We are pleased to hear that you found the paper logically structured and well-written. Our aim was to provide a comprehensive overview of the genetic background and treatment options for HS, and we are glad to see that our efforts have been recognized.
- In my opinion, some drugs currently used off-label or under development are missing. You can look at recent reviews on the topic and integrate:
- Świerczewska Z, Lewandowski M, Surowiecka A, Barańska- Rybak W. Immunomodulatory Drugs in the Treatment of Hidradenitis Suppurativa Possibilities and Limitations. Int J Mol Sci. 2022 Aug 26;23(17):9716. doi: 10.3390/ijms23179716 . PMID: 36077114 ; PMCID: PMC9456321 .
- Ocker L, Abu Rached N, Seifert C, Scheel C, Bechara FG. Current Medical and Surgical Treatment of Hidradenitis Suppurativa-A Comprehensive Review. J Clin Med. 2022 Dec 6;11(23):7240. doi: 10.3390/jcm11237240 . PMID: 36498816 ; PMCID:PMC9737445 .
Response: Thank you for your valuable suggestion regarding the inclusion of some drugs currently used off-label or under development. After carefully reviewing the provided literature and conducting additional searches for related articles, we have incorporated information on three drugs targeting TNF-alpha, namely etanercept, golimumab, and certolizumab. For IL-17 inhibitors, we have added details on ixekizumab, sonelokimab, and izokibep. In the section covering IL-1 inhibitors, anakinra and canakinumab have been included, discussing their current off-label usage or potential in development. Additionally, we have cited the two references you provided in the reference section of the manuscript. I hope these additions enhance the comprehensiveness and relevance of the review.
Reviewer 2 Report
Comments and Suggestions for Authors
It is my great pleasure to have an opportunity to review this interesting article. The authors discussed the mechanism based on the genetic and molecular pathogenesis and treatment options of HS. This paper is well-written and very interesting. I have only a couple of comments.
1. The authors described that JAK-STAT signaling can contribute to the pathogenesis of HS in sections 2.3 and 2.5. Some literature reported the efficacy and safety of JAK inhibitors for HS. Those should be added to section 3.
2. Figure 1 presents immunological mechanisms and current biologic agents of HS. It is well organized. However, the quality of the figure is very low and looks hand-drawn. It should be modified to be a digital illustration.
Author Response
Response to reviewer 2
It is my great pleasure to have an opportunity to review this interesting article. The authors discussed the mechanism based on the genetic and molecular pathogenesis and treatment options of HS. This paper is well-written and very interesting. I have only a couple of comments.
Response: Thank you for taking the time to review our article. We are delighted that you found the discussion on the genetic and molecular pathogenesis, as well as treatment options for HS, interesting and well-written. Your positive feedback is greatly appreciated.
- The authors described that JAK-STAT signaling can contribute to the pathogenesis of HS in sections 2.3 and 2.5. Some literature reported the efficacy and safety of JAK inhibitors for HS. Those should be added to section 3.
Response: Thank you for your suggestion. We appreciate your recommendation to include information on the efficacy and safety of JAK inhibitors in the treatment of HS in the "Current and Potential Therapeutic Agents Targeting Immune Mediators in Hidradenitis suppurativa" section. After an extensive literature search and careful consideration, we have added a new paragraph, 4.5. Janus Kinase Inhibitors, to introduce the findings related to JAK inhibitors. This includes information about ongoing clinical trials, suggesting a promising future for the treatment of HS with these inhibitors. We trust that this addition provides a more comprehensive overview of therapeutic agents targeting immune mediators in HS.
- Figure 1 presents immunological mechanisms and current biologic agents of HS. It is well organized. However, the quality of the figure is very low and looks hand-drawn. It should be modified to be a digital illustration.
Response: We appreciate your feedback regarding the quality of Figure 1. Following your suggestion, I have diligently worked on refining and enhancing its visual presentation. The adjustments made are aimed at improving the overall drawing quality to align better with the academic style, making it appear more professional and less hand-drawn.

Reviewer 3 Report
Comments and Suggestions for Authors
dear authors, your paper is a valuable review of pathogenetic mechanism of HS, from a genetic point of view. Also, molecular therapies are herein discussed.
I suggest to add a methodological paragraph in which you explain how this review has been realized (keywords etc)
in the introduction you should add diagnostic tools for HS, i.e. ultrasound and diagnostic criteria and staging methods (papers by wortsman x and nazzaro g). I suggest also to refer to sebocystomatosis as a genetic inherited disease, clinically similar to HS.
Author Response
Response to reviewer 3
Dear authors, your paper is a valuable review of pathogenetic mechanism of HS, from a genetic point of view. Also, molecular therapies are herein discussed.
Response: Thank you for your thoughtful review of our paper. We appreciate your acknowledgment of our efforts to provide a valuable review of the pathogenetic mechanism of HS, particularly focusing on the genetic perspective. Additionally, we are pleased to hear that the discussion on molecular therapies has been recognized.
- I suggest to add a methodological paragraph in which you explain how this review has been realized (keywords etc).
Response: Thank you for your recommendation to include a methodological paragraph. We have addressed this suggestion by incorporating a new section, 2. Search Strategy, in the manuscript. This section provides a detailed description of the research methodology employed, including the keywords used during the literature search. We believe that this addition enhances the transparency of the review process.
- In the introduction you should add diagnostic tools for HS, i.e. ultrasound and diagnostic criteria and staging methods (papers by wortsman x and nazzaro g).
Response: Thank you for your comment regarding the inclusion of information on HS diagnostic tools, diagnostic criteria, and disease staging in the introduction. We have incorporated the recommended content into the introduction, highlighted with a yellow background. One paragraph now covers information related to the diagnosis of HS, while another paragraph provides an introduction to disease staging. Additionally, we have cited the references you provided in the reference section of the manuscript.
- I suggest also to refer to sebocystomatosis as a genetic inherited disease, clinically similar to HS.
Response: Thank you for your suggestion. After conducting a thorough search and reviewing relevant literature, we have added a description highlighting the similarities between steatocystoma multiplex and HS. These additions are marked with a yellow background in the introduction, specifically in the section discussing symptoms of HS.

Round 2
Reviewer 1 Report
Comments and Suggestions for Authors
Dear Authors,
the revisions you provided increased the value of your manuscript. I have no other comments from my side.